# Range Gate Pull-Off Mainlobe Jamming Suppression Approach with FDA-MIMO Radar: Theoretical Formalism and Numerical Study

Pengfei Wan [1,2], Yuanlong Weng [1,3], Jingwei Xu [1,*] and Guisheng Liao [1]

1   National Lab of Radar Signal Processing, School of Electronic Engineering, Xidian University,
    Xi'an 710071, China; wanpengfei@stu.xidian.edu.cn (P.W.); 21023110385@stu.xidian.edu.cn (Y.W.);
    gsliao@xidian.edu.cn (G.L.)
2   Air and Missile Defence College, Air Force Engineering University, Xi'an 710051, China
3   The 38th Research Institute of China Electronics Technology Group Corporation, Hefei 230088, China
*   Correspondence: jwxu@xidian.edu.cn; Tel.: +86-1819-2039-179

**Abstract:** With the development of an electronic interference technique, the self-defense jammer can generate mainlobe jamming using the range gate pull-off (RGPO) strategy, which brings serious performance degradation of target tracking for the ground-based warning radar. In this paper, a RGPO mainlobe jamming suppression approach is proposed, with a frequency diverse array using multiple-input multiple-output (FDA-MIMO) radar. The RGPO mainlobe jamming differs from the true target in slant range, thus it is possible to identify the true target from the RGPO mainlobe jammings by exploiting the transmit beampattern diversity of FDA-MIMO radar. A RGPO mainlobe jamming suppression approach is devised by using joint transmit–receive beamforming for a group of range sectors. The jamming suppression performance is studied, in consideration of practical time-delay of RGPO jamming. Simulation examples are provided to verify the effectiveness of the proposed approach.

**Keywords:** ground-based warning radar; mainlobe jamming suppression; range gate pull-off; frequency diverse array; multiple-input multiple-output radar





## 1. Introduction

Ground-based warning radar plays an important role in national defense applications [1,2]. With the low sidelobe antenna technique and large time-bandwidth products, ground-based warning radar provides sufficient anti-jamming ability against the sidelobe jammers. However, with the development of electronic interference techniques in the advanced weapons, ground-based warning radar encounters extremely hostile environments in the mainlobe region [3–6], which becomes a great challenge for the traditional phased array radar systems.

It is known that mainlobe jamming is not easy to implement. Nevertheless, the self-defensive jammer is well-known equipment for generating mainlobe jamming, whose direction is exactly the same as that of the true target [7–11]. Commonly, the jammer intercepts the radiated waveform of radar and performs multi-dimensional modulation in the range, speed, power, etc. With the multi-dimensional modulation, it is possible to generate many kinds of deceptive jamming signals, also referred to as false targets, which leads to errors in the radar by mistakenly tracking the false targets. Therefore, deceptive jamming has serious consequences in regard to the radar system, such as increased false alarms, missing of true targets, and extremely heavy computational burden. Among the many flexible jamming modulation techniques, range gate pull-off (RGPO) is an efficient one [12–15]. It uses time-delay modulation in fast-time domain, and it is possible to generate many overlapped false targets. In [12], an optimal multi-frame RGPO jamming strategy

is proposed, based on black-box optimization idea, using a particle swarm optimization algorithm. In [14], the phase quantization and RGPO delay quantization of the jamming signals emitted by electronic counter–countermeasures are studied. It pointed out that the spectrum of jamming signal presents a primary term and false terms after phase quantization, while GPRO introduces negligible distortion. There are also some refine approaches to improve the effectiveness of RGPO. A bidirectional RGPO jamming is proposed in [15], which can resist the pulse leading edge or trailing edge tracking technique.

At present, anti-deceptive jamming approaches in monostatic radar, such as waveform diversity [16,17], kinematics information [18,19], and polarization information [20,21], have been utilized to extract information from jamming background. The approach, based on pulse agility, may cause high-range sidelobe and increase the time complexity. Focus on the difference of scattering characteristics between target and jamming, a strategy to identify true and false targets in data domains, is proposed in [18]. However, techniques based on polarization filtering in [20] will become invalid when the polarization mode of jammer is also variable, which requires further considerations in practice.

In recent years, a flexible beam scanning array, referred to as frequency diverse array, (FDA) was first introduced by Antonik [22]. The FDA differs from its traditional phased array counterpart because the carrier frequency across the array elements is increased; thus, it generates range/time-angle-dependent beampattern in the space [23,24]. In order to improve the target's indication ability in the range and angle domains, a dot-shaped, range–angle beampattern method was proposed in [25]. A subarray-based method [26] and grid-less compressed sensing-based algorithm [27] were proposed, based on FDA, for range and angle parameters estimation. The multiple-input multiple-output (MIMO) technique is combined with FDA to generate flexible control of transmit degree of freedom (DOF) in [28,29]. Furthermore, the range–angle-dependent beampattern of FDA is utilized to handle the repeated mainlobe jamming suppression issue with FDA-MIMO radar in [30–34]. The principle of FDA-MIMO, for distinguishing between true target and deceptive jamming signals, is first established in [30]; however, the deceptive jamming model misses the time-delay modulation within the jammer. In [33], a robust deceptive jamming suppression method, based on covariance matrix reconstruction, is proposed to handle the practical errors. The deceptive jamming model is revised in [34] by considering the time-delay modulation in the jammer, which is also pointed out in [31]. With the corrected signal and jamming model in the FDA-MIMO framework, it is reported that *a priori* knowledge of true target, such as estimated target range and angle parameters, is required for the suppression of deceptive jamming [24,34,35]. Besides, the jamming signal can be well-suppressed, under the condition that its time-delay is larger than one pulse repetition time (PRT). Transceive beamforming methods, with accurate nulling in the joint range and angle domain, were presented to suppress the range–angle sidelobe interference effectively in [36]. Moreover, the non-adaptive beampattern of FDA-MIMO radar is sophisticated designed to cope with practical estimation errors, such as angle mismatch, discretized range error, and array element response error in [37]. In [38], the anti-jamming ability was improved by joint optimization of polarization and frequency step. On the other side, the jamming design method is also studied in the literature. In [39], a deceptive jamming approach against FDA radar is proposed by utilizing the beam gain to avoid being nulled.

In this paper, a RGPO mainlobe jamming suppression approach is proposed with FDA-MIMO radar, which improves the anti-jamming ability for ground-based warning radar systems. The RGPO jamming strategy is studied, and its practical constraint on range is utilized. The RGPO false targets, with large and small time-delays, are both considered in this paper, which is different from the existing works in literature [30–37]. To maintain leading edge tracking performance, an enlarged range windowing strategy is used to cover a relatively large range region for target confirmation. Within the large range region, the two-dimensional adaptive beamforming in the joint transmit–receive spatial frequency is applied for a group of range sectors. These range sectors are collected in a group for

confirmation of the leading edge. Therefore, the proposed method can be applied in the situation with very fast, repeated false targets.

The remainder of this paper is organized as follows. Section 2 presents the signal model and the range–angle-dependent transceive beampattern in FDA-MIMO. The algorithm to suppress RGPO mainlobe jamming within multiple range sectors is explored in Section 3. Numerical results in Section 4 are provided to verify the performance of the proposed methods.

*Notations*: Boldface is used for vectors $x$ (lower case), whose $n$-th entry is $[x]_n$, with matrices A (upper case). The transpose and conjugate transpose operators are denoted by the symbols $(\cdot)^T$ and $(\cdot)^H$, respectively. $C^{N \times 1}$ and $C^{N \times M}$ are, respectively, the sets of $N$-dimensional vectors of complex numbers and $N \times M$ complex matrices; $\odot$ and $\otimes$ represent the Hadamard product and the Kronecker product, respectively. The letter $j$ represents the imaginary unit (i.e., $j = \sqrt{-1}$); $[a, b]$ indicates a closed interval in real number space. Finally, max$\{\cdot\}$ and min$\{\cdot\}$ denote the maximum and minimum values within the feasible set.

## 2. Signal Model of FDA-MIMO Radar

It is assumed that the FDA-MMO radar system is a linear array, composed of $M$ transmit antenna and $N$ receive antenna elements. Assume that the transmit and receive antenna elements are omni-directional, identical, and uniform. The signal of $m$-th transmit element can be expressed as:

$$s_m(t) = \text{rect}\left(\frac{t}{T_p}\right)\Phi_m(t)\exp\{j2\pi f_m t\} \tag{1}$$

where $t$ is the time variable, $\text{rect}\left(\frac{t}{T_p}\right) = \begin{cases} 1, 0 \leq t \leq T_P \\ 0, else \end{cases}$ is the pulse function, $\Phi_m(t)$ is the baseband modulation signal corresponding to the $m$-th transmit element, $f_m = f_0 + (m-1)\Delta f$ is the transmit frequency corresponding to $m$-th transmit element, $f_0$ is the reference carrier frequency, and $\Delta f$ is the frequency increment.

Assume a target with range and angle parameters of $(R,\theta)$. Then, the echo signal corresponding to the $m$-th transmit and $n$-th receive elements can be expressed as:

$$x_{s,m,n}(t - \tau_{m,n}) = \beta_{s0}\text{rect}\left(\frac{t - \tau_{m,n}}{T_p}\right)\Phi_m(t - \tau_{m,n})\exp\{j2\pi f_m(t - \tau_{m,n})\} \tag{2}$$

where $\beta_{s0}$ represents the complex coefficient of the target echo signal, and $\tau_{m,n}$ is the time-delay corresponding to the transmit–receive pair, which is expressed as

$$\tau_{m,n} = \tau_0 - \frac{d(m-1)\cos(\theta)}{c} - \frac{d(n-1)\cos(\theta)}{c} \tag{3}$$

where $\tau_0$ is the delay, due to common propagation, and $d$ is the element spacing of the transmit and receive antenna array. Under the far-field source and narrowband assumption, the echo signal can be approximately written as

$$x_{s,m,n}(t - \tau_0) \approx \beta_{s0}\text{rect}\left(\frac{t - \tau_0}{T_p}\right)\Phi_m(t - \tau_0)\exp\{j2\pi\Delta f(m-1)(t - \tau_{m,n})\}\exp\{j2\pi f_0(t - \tau_{m,n})\} \tag{4}$$

Thus, the target signal received by the $n$-th receive element is summation of all echoes corresponding to $M$ transmit elements, which is written as:

$$\begin{aligned} x_{s,n}(t - \tau_0) &= \sum_{m=1}^{M} x_{s,m,n}(t - \tau_0) \\ &\approx \sum_{m=1}^{M} \beta_{s0}\text{rect}\left(\frac{t - \tau_0}{T_p}\right)\Phi_m(t - \tau_0)\exp\{j2\pi\Delta f(m-1)(t - \tau_{m,n})\}\exp\{j2\pi f_0(t - \tau_{m,n})\} \end{aligned} \tag{5}$$

In the receiver, the echo signal is down-converted to baseband, multi-waveform separated, and rearranged as a vector in receive-wise form. Thus, we can obtain:

$$s(t) = \beta_s \delta(t - \tau_0) a(\tau_0, \theta) \otimes b(\theta) \tag{6}$$

where $\beta_s$ is the complex coefficient of the target echo after pulse compression, $\delta(t - \tau_0)$ is the sinc function, indicating that the target is associated with time-delay, $\tau_0$, $\otimes$ is the Kronecker product, and $a(\tau_0, \theta) \in \mathbb{C}^{M \times 1}$ and $b(\theta) \in \mathbb{C}^{N \times 1}$ are, respectively, the transmit and receive steering vectors of the target, they are written as:

$$
\begin{aligned}
a(\tau_0, \theta) \quad &= a_r(\tau_0) \odot a_\theta(\theta) \\
&= [1, \exp(-j2\pi\Delta f \tau_0), \ldots, \exp(-j2\pi\Delta f \tau_0 (M - 1))]^T \\
&\odot \left[1, \exp\left(j2\pi\frac{d\cos(\theta)}{\lambda}\right), \ldots, \exp\left(j2\pi\frac{d(M-1)\cos(\theta)}{\lambda}\right)\right]^T \\
&= \left[1, \exp\left\{-j2\pi\Delta f \tau_0 + j2\pi\frac{d}{\lambda}\cos(\theta)\right\}, \cdots, \exp\left\{-j2\pi\Delta f \tau_0 (M - 1) + j2\pi\frac{d}{\lambda}(M - 1)\cos(\theta)\right\}\right]^T
\end{aligned} \tag{7}
$$

$$
b(\theta) = \left[1, \exp\left\{j2\pi\frac{d}{\lambda}\cos(\theta)\right\}, \cdots, \exp\left\{j2\pi\frac{d}{\lambda}(N - 1)\cos(\theta)\right\}\right]^T \tag{8}
$$

where $\odot$ denotes the Hadamard product,

$$
a_r(\tau_0) = [1, \exp(-j2\pi\Delta f \tau_0), \ldots, \exp(-j2\pi\Delta f \tau_0 (M - 1))]^T
$$

and

$$
a_\theta(\theta) = \left[1, \exp\left(j2\pi\frac{d\cos(\theta)}{\lambda}\right), \ldots, \exp\left(j2\pi\frac{d(M - 1)\cos(\theta)}{\lambda}\right)\right]^T
$$

are, respectively, the range and angle steering vectors. As can be seen from (7), the range steering vector of FDA-MIMO radar contains both the time-delay and angle information. Therefore, FDA-MIMO radar has the ability to distinguish targets with different time-delays, which provides additional degree of freedom (DOF) in jamming suppression.

Assume the airborne jammer device intercepts radar waveform and releases RGPO jamming signals. Here, we consider the self-defense jammer with range and angle parameters the same as the true target. Thus, for the $p$-th false target, the jamming signal corresponding to the $m$-th transmit element and $n$-th receive element can be expressed as:

$$
x_{j,p,m,n}(t - \tau_{0p}) \approx \beta_{j,p}\mathrm{rect}\left(\frac{t - \tau_{0p}}{T_p}\right)\Phi_m(t - \tau_{0p})\exp\{j2\pi\Delta f(m - 1)(t - \tau_{p,m,n})\}\exp\{j2\pi f_0(t - \tau_{p,m,n})\} \tag{9}
$$

where $\tau_{0p} = \tau_0 + \Delta\tau_p$ is the equivalent time-delay of the $p$-th jamming signal, $\Delta\tau_p$ is the time-delay of the $p$-th false target within the jammer, $\tau_0$ is associated with time-delay of the true target, and

$$
\tau_{p,m,n} = \tau_{0p} - \frac{d(m - 1)\cos(\theta)}{c} - \frac{d(n - 1)\cos(\theta)}{c}
$$

is the total time delay of the $p$-th false target corresponding to different transmit–receive pair. Assume that the self-defense jammer generates $P$ false targets, thus, the deceptive jamming signal can be expressed as:

$$
i(t) = \sum_{p=1}^{P} \beta_p \delta(t - \tau_{0p}) a(\tau_{0p}, \theta) \otimes b(\theta) \tag{10}
$$

where $i(t) \in \mathbb{C}^{MN \times 1}$. It is seen that the repeated RGPO jamming signal has the same form, with the true target signal. However, as the transmit steering vector contains time-delay information, and the true and false targets differ from each other by the time-delay

parameter. Considering the true target, repeated RGPO jamming, and noise components, the echo is expressed as:

$$
\begin{aligned}
\boldsymbol{x}(t) &= \boldsymbol{s}(t) + \boldsymbol{i}(t) + \boldsymbol{n}(t) \\
&= \beta_s \delta(t - \tau_0) \boldsymbol{a}(\tau_0, \theta) \otimes \boldsymbol{b}(\theta) + \sum_{p=1}^{P} \beta_p \delta(t - \tau_{0p}) \boldsymbol{a}(\tau_{0p}, \theta) \otimes \boldsymbol{b}(\theta) + \boldsymbol{n}(t)
\end{aligned}
\tag{11}
$$

where $\boldsymbol{n}(t) \in \mathbb{C}^{MN \times 1}$ is complex Gaussian white noise.

## 3. Mainlobe RGPO Jamming Suppression for FDA-MIMO Radar

In this section, the general mechanism of RGPO jamming is discussed, and the difference between the true target and RGPO jamming is analyzed. Usually, the radar tracks a particular target within a limited range region. Therefore, the RGPO might cause severe performance degradation for the radar. In this paper, a RGPO mainlobe jamming suppression approach is devised by using an enlarged range windowing strategy.

In the jammer side, the electronic support system usually starts working after confirmation of being tracked by a hostile radar. The self-defense jammer can generate a great amount of false targets without specific modulation. It can also generate some particularly designed false targets for confusion. Generally, RGPO can be categorized into front pull-and back pull-off, according to different time delays. For the front pull-off RGPO jamming, the time delay within the jammer gradually decreases, and it results in false targets, approaching the radar faster than the true target. In this case, the time delay is as large as pulse repetition interval (PRI) or several times of PRI. For the back pull-off RGPO jamming, the time delay within the jammer gradually increases, and it results in false targets departing the radar faster than the true target. It is reported that it is difficult to handle the back pull-off RGPO jamming because the time-delay false targets are within the same pulses. In this paper, the enlarged range windowing strategy provides a feasible solution for the confirmation of the true target under the RGPO mainlobe jamming environment.

Recall (9) of the equivalent time-delay of the *p*-th jamming signal and pull-off time-delay varies, with respect to slow time or pulse. The discrete time-delay within the jammer is rewritten as:

$$
\tau_0(k) = \tau_0 + \Delta\tau(k)
\tag{12}
$$

where $k$ denotes the pulse index number. Here, we omit the false target indicator $p$ for simplicity. For the constant velocity pull-off strategy, the time delay satisfies the following relation:

$$
\Delta\tau(k) = (q-1)T_r + \frac{2v_a T_r}{c}(k-1)
\tag{13}
$$

where $T_r$ is the PRI, $q$ is the number of delayed pulse, and $v_a$ is the relative velocity of the false target. If $v_a > 0$, the false targets are pulling off backwards, while, if $v_a < 0$, the false targets are pulling off forwards. It is pointed that the modulation function of the time-delay determines the motion model of false target. The purpose of RGPO is to guide the radar to the false target. Generally, the absolute range difference between the true and false targets gradually increases, as shown in Figure 1. With the increment of range difference between the true and false targets, the tracking gate of radar might be pulled to another range, far from the true target.

As aforementioned in the FDA-MIMO radar, the transmit steering vector in (7) is dependent on the time-delay and angle parameters, which can be utilized to distinguish true and false targets. The *m*-th entry of transmit steering vector in (7) can be expressed as:

$$
[\boldsymbol{a}(\tau_0, \theta)]_m = \exp\left(-j2\pi\Delta f\tau_0(m-1) + j2\pi\frac{d\cos(\theta)}{\lambda}(m-1)\right)
\tag{14}
$$

where $[a]_m$ denotes the $m$-th entry of the steering vector. The equivalent transmit spatial frequency can be written as:

$$f_T(\tau_0, \theta) = -\Delta f \tau_0 + \frac{d \cos(\theta)}{\lambda} \tag{15}$$

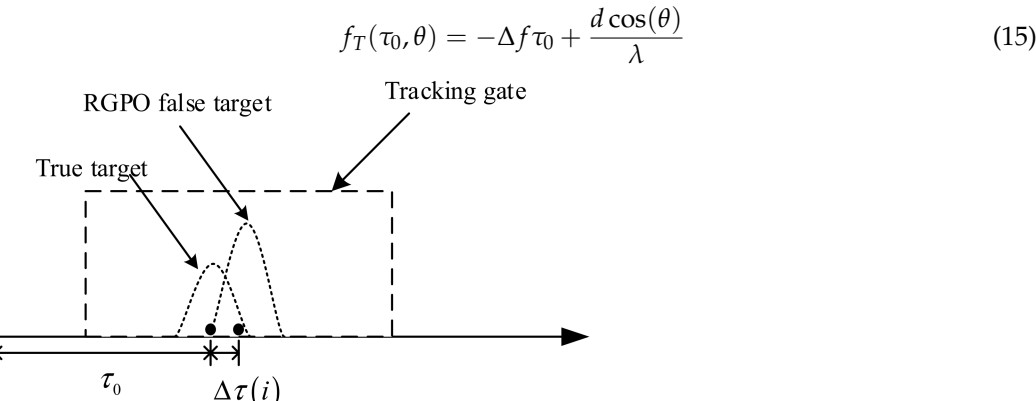

**Figure 1.** Schematic of RGPO false target.

This is the transmit spatial frequency of true target. Similarly, the transmit spatial frequency of false target can be expressed as:

$$f_T(\tau_0 + \Delta\tau, \theta) = -\Delta f(\tau_0 + \Delta\tau) + \frac{d \cos(\theta)}{\lambda} \tag{16}$$

Thus, the true and false targets differ from each other by the transmit spatial frequency. The difference can be obtained as:

$$\delta f_T(\tau_0, \theta) = f_T(\tau_0 + \Delta\tau, \theta) - f_T(\tau_0, \theta) = -\Delta f \Delta\tau \tag{17}$$

As the receive spatial frequency is only dependent on the angle parameter, the true and false targets are the same in the receive spatial frequency, which can be written as:

$$f_R(\theta) = \frac{d \cos(\theta)}{\lambda} \tag{18}$$

Combing the transmit and receive spatial frequency, it is capable for FDA-MIMO radar to distinguish the true and target in the joint transmit–receive spatial frequency domain. It is seen that they are separated, due to the different of transmit spatial frequency, which is superior, compared with its phased array radar counterpart. It is noted that the secondary range dependence compensation is proposed in [28], in order to remove the range dependence of the transmit spatial frequency. Therefore, the spectrum position of true target will appear on the diagonal line in Figure 2. It is verified that the true and forward pull-off RGPO false targets are easily distinguishable, due to large time-delay into the next pulses. In this situation, the false targets are associated with a different pulse index, and they have different spectrum positions in joint transmit–receive spatial frequency domain after secondary range dependence compensation. In contrast, it is difficult to distinguish the true target and pull-off RGPO false target with small time-delay [34]. In this situation, the true and false targets will be very close in the joint transmit–receive spatial frequency domain after secondary range dependence compensation. In the following, an enlarged range windowing strategy is adopted to handle this issue.

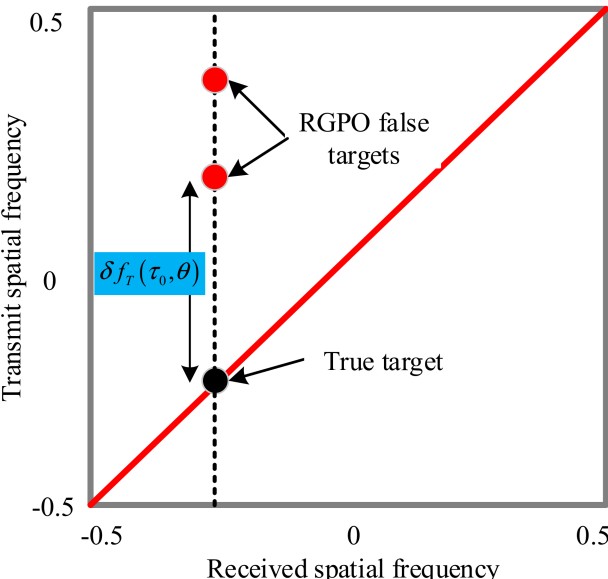

**Figure 2.** Spectrum position of true and false targets in joint transmit–receive spatial domain.

In this paper, an enlarged range windowing strategy is devised to cover a relatively large range region for target confirmation. Besides, multiple range sectors are grouped together for simultaneous target confirmation. Therefore, the two-dimensional adaptive beamforming in joint transmit–receive spatial frequency is updated to cover a group of range sectors. For a given range sector, the beamformer can be formulated as:

$$
\begin{cases}
\min w_g^H R w_g \\
s.t. w_g^H (a(\tau_g, \theta) \otimes b(\theta)) = 1
\end{cases}, \theta = \theta_0, \tau_g \in \tau_0 + [\xi_1, \xi_2], g = 1, \cdots, G \tag{19}
$$

where $R$ is the covariance matrix, $w_g$ is the $g$-th weight vector in the group, $\theta$ is the presumed angle of true target, $\xi_1$ and $\xi_2$ are the lower and upper bounds, and $G$ is the total number of two-dimensional adaptive beamformer for the enlarged range region. In other words, we apply several adaptive beamforming for the interested range region. Thus, the covered range region increases when tracking the target. It should be pointed out that a series of weight vectors, calculated with this grouped range sectors, is different from the robust adaptive beamforming techniques using multiple uncertainty sets [40]. This strategy is helpful for protecting the true target from being pulled-off. In the practical tracking stage of radar system, the target information can be predicted using prior information in the previous working period. However, the tracking range gate in the conventional radar is related to the range gate size and tracking accuracy. The small tracking range gate is not appropriate for the dense deceptive jamming environment. In contrast, this paper provides an enlarged range windowing strategy by using two-dimensional adaptive beamforming for a group of range sectors, which enables dynamically adjusted range, gated during the subsequent tracking procedure. The proposed procedure is plotted Figure 3. The input data is processed with multiple waveform separation to obtain the transmit DOF and secondary range dependence compensation (SRDC), in order to remove the range dependence. In the sequel, a group of two-dimensional adaptive beamforming is applied to suppress those false targets delayed to the next pulses. The true target can be abstracted with leading edge confirmation within the enlarged range region.

The covariance matrix can be estimated using the received data [34] and constructed with prior knowledge [36]. The estimated covariance matrix is written as:

$$
\hat{R} = E\left\{ x(t) x^H(t) \right\} = \sum_{l=1}^{L} x_l x_l^H \tag{20}
$$

where $E\{\cdot\}$ is the expectation operator. The estimated covariance matrix might be contaminated by the true target, if the data under test is included. Besides, the covariance matrix might be under-determined because the RGPO false targets are discretely and randomly distributed in the range dimension. Some specific methods are proposed to improve the estimation accuracy of the covariance matrix [35]. Nevertheless, as the transmit spatial frequency of the power spectrum of the true and false targets in the joint transmit–receive spatial frequency domain are dependent on the frequency increment. By properly choosing the system parameters, the position of false target can be predicted, providing the angle parameter. In this case, the covariance matrix is constructed as:

$$\tilde{\boldsymbol{R}} = \sum_{\Delta\theta \in \Theta} \sum_{\tau} \boldsymbol{v}(\tau, \theta + \Delta\theta) \boldsymbol{v}^H(\tau, \theta + \Delta\theta) \tag{21}$$

where $\theta$ is within a small uncertainty set, due to the estimation error, $\Delta\theta$ is the uncertainty set parameter, and $\boldsymbol{v}(\tau, \theta + \Delta\theta) = \boldsymbol{a}(\tau, \theta + \Delta\theta) \otimes \boldsymbol{b}(\theta + \Delta\theta)$ is the steering vector of the predicted false target. The summation of the covariance matrix within the uncertainty set can enhance the robustness of the beamformer. After SRDC, the steering vector of the false targets are discretely positioned in the joint transmit–receive spatial frequency domain. The corresponding time-delay parameter degenerates to range ambiguous region index.

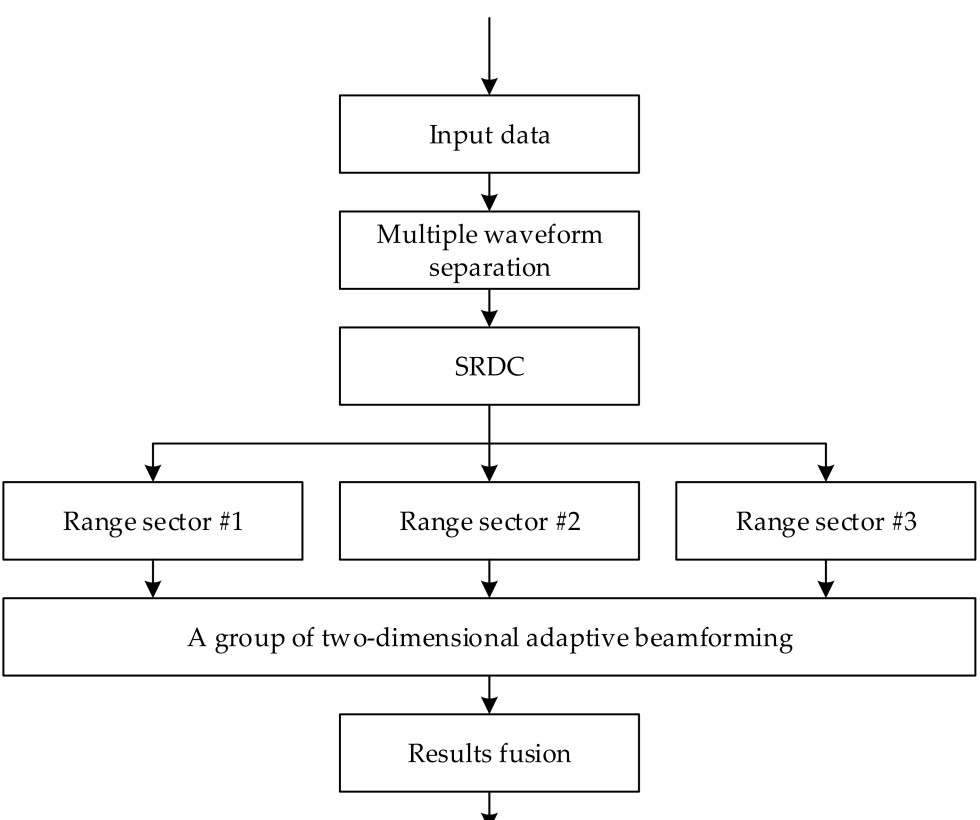

**Figure 3.** Simultaneous tracking of a group of range sectors.

The difference between true and false targets becomes larger, with respect to time. Thus, the lower and upper bounds within the interested range region vary during the tracking period. Specifically, the lower and upper bounds are written as:

$$\begin{aligned} \xi_1 &= \min\{\tau_1, \tau_2, \cdots, \tau_G\} - \varepsilon \\ \xi_2 &= \max\{\tau_1, \tau_2, \cdots, \tau_G\} + \varepsilon \end{aligned} \tag{22}$$

where $\tau_1, \tau_2 \cdots \tau_G$ is the discretized range estimates of the true and false targets, corresponding to the group of beamformer, and $\varepsilon$ is the tolerance range deviation for user choice. With

the proposed scheme, it is possible to obtain the leading edge of the interested range region and, thus, protect the true target in the tracking period. Notice that the two-dimensional transmit–receive beamforming with FDA-MIMO will cause a global noise level increment, which requires further work.

## 4. Simulations

In the section, the simulation examples are provided to verify the effectiveness of the proposed scheme. In the simulation, the transmit and receive arrays are both a uniform linear array with half-wavelength spaced. Both transmit and receive arrays contain 10 elements. The carrier frequency is 10 GHz. The frequency increment is 21 kHz and PRF is 10 kHz. We assume 16 coherent pulses are collected within the coherent processing interval. The sample frequency is 10 MHz and corresponding range sample number is 1000 in the range dimension. The true target is positioned at range 10 km, with an angle of 0 degree and signal-to-noise-ratio (SNR) of 10 dB. Without specifying otherwise, the jamming-to-noise-ratio (JNR) of RGPO jamming is 15 dB. Besides, the suppressive jamming is also considered in the simulation, with an angle of 30 degrees, as well as JNR 20 dB.

### 4.1. Property Analysis of RGPO Jamming

The properties of the receive signals, Figure 4. The Fourier power spectrum is calculated in this example. As aforementioned, the four-dimension data cube, i.e., receive–transmit Doppler range dimension, is obtained after multi-waveform separation. It is known that the suppressive jamming is approximately white, distributed in normalized Doppler frequency and range dimensions, which can be seen from the Fourier power spectrum in Figure 4a. The RGPO false targets are buried in the suppressive jamming because the deceptive jamming has lower power than the suppressive jamming. It is seen from the power spectrum in the receive spatial angle and range dimensions Figure 4c that the suppressive jamming is focused at an angle of 30 degrees, while the RGPO false targets are from 0 degree. Besides, it is seen that the RGPO false targets are randomly distributed in range dimension. In this simulation, we assume that the 200 RGPO false targets are randomly distributed within the subsequent PRI. From the power spectrum in transmit spatial angle and range dimensions Figure 4b, it is also indistinctly seen that the distribution of the false targets is the transmit angle and range-dependent. In order to clearly show the properties of RGPO false targets, the Fourier power spectrum of the received signal without the suppressive jamming is provided in Figure 4, as a comparison. It is seen from Figure 4a that the that RGPO false target are also randomly positioned in range and Doppler dimensions. Due to the introduction of frequency diversity, the distribution of false targets in the range and transmit angle dimensions are coupled, which can be obviously observed from Figure 4b. However, these false targets are focused in degree 0 of the receive dimension, as shown in Figure 4c.

In Figure 5, the Capon power spectra, before and after SRDC, in the joint transmit–receive spatial domain are provided. It is seen that the suppressive jamming is focused at an angle of 30 degrees in the receive dimension and non-focused in the transmit dimension, whether before or after SRDC. For the RGPO false targets, it is seen that they are randomly distributed in the transmit dimension before SRDC in Figure 5a. However, they become focused after SRDC in Figure 5b. Note that these false targets are associated with having 0 degree in the receive dimension, which is the same angle as the true target. In the simulation, these false targets are delayed within a CPI. Thus, they might be within the same pulse as the true target if the delay is small, and they might also belong to the next pulse if the delay is large enough. Therefore, there are two positions corresponding to these false targets in the joint transmit–receive spatial domain, as seen from Figure 5b.

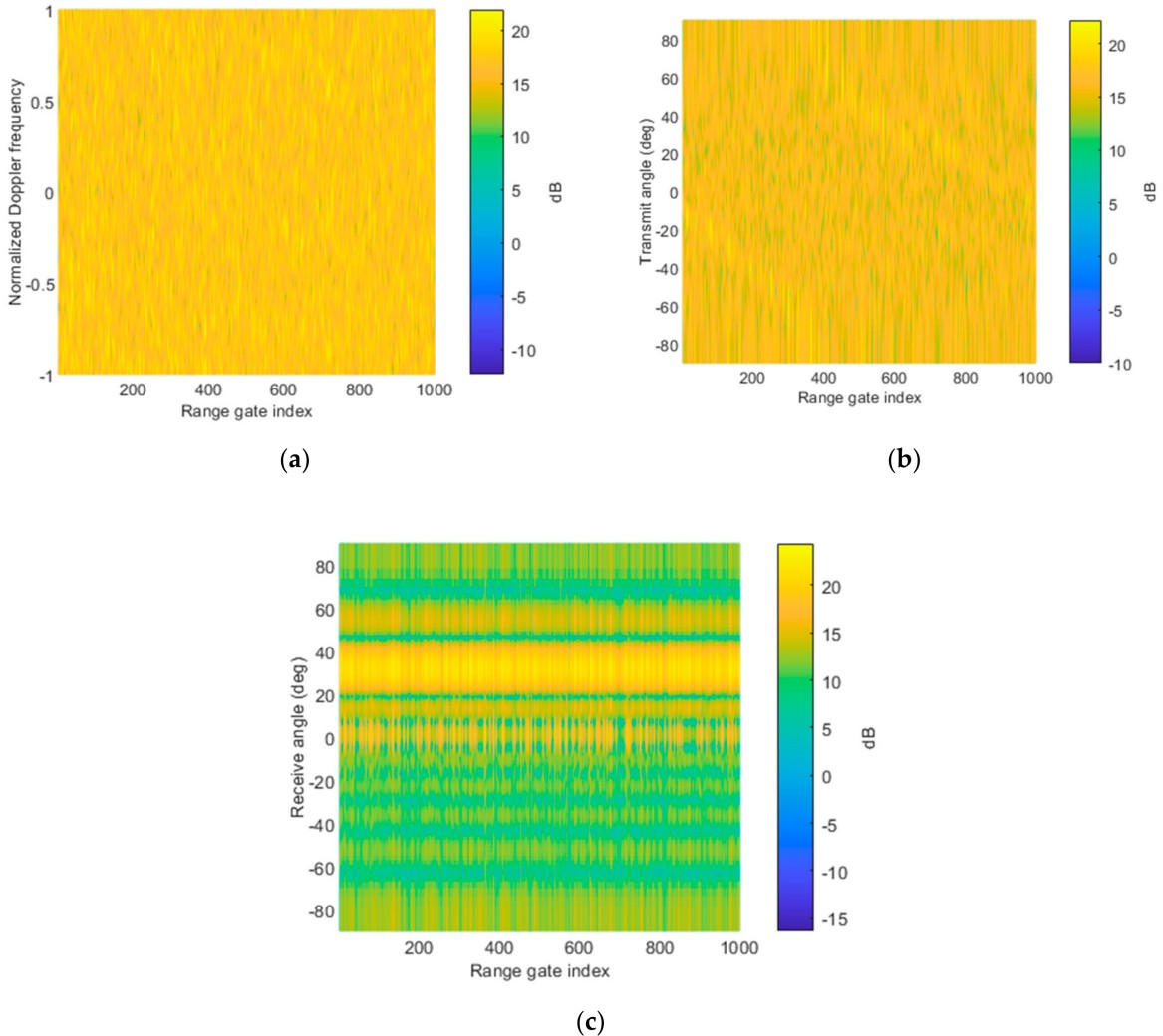

**Figure 4.** Fourier power spectrum of receive signal, including the true target, suppressive jamming, and RGPO false targets in the (**a**) normalized Doppler frequency and range dimensions; (**b**) transmit spatial angle and range dimensions; and (**c**) receive spatial angle and range dimensions.

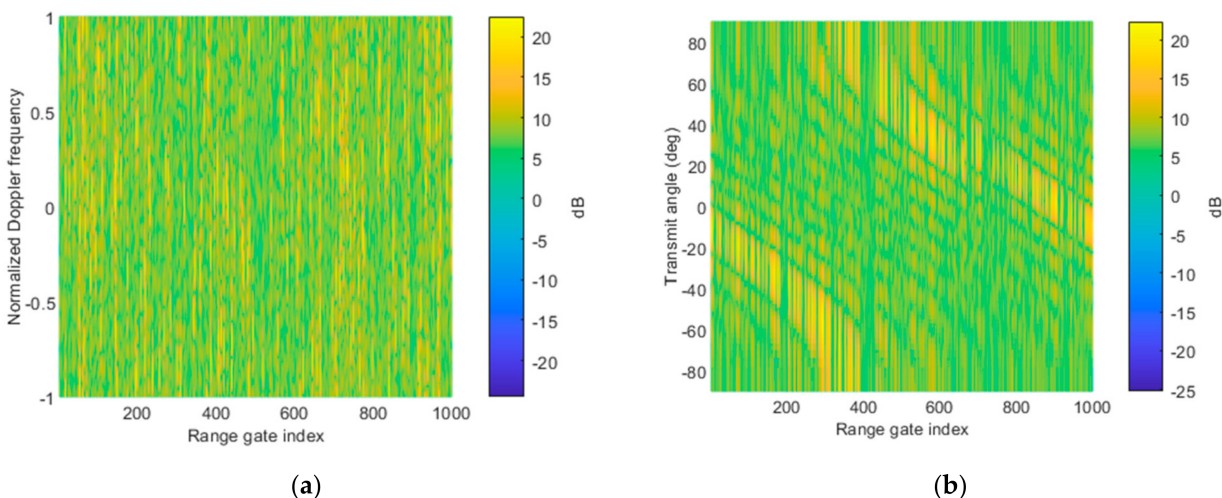

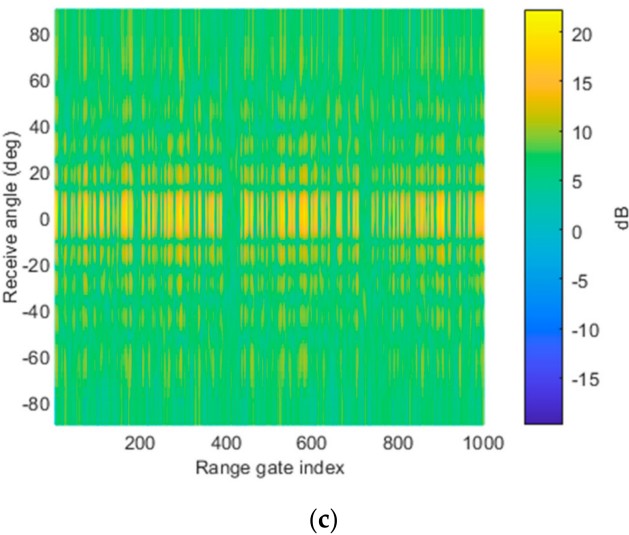

(**c**)

**Figure 4.** Fourier power spectrum of the receive signal, including the true target and RGPO false targets in the (**a**) normalized Doppler frequency and range dimensions; (**b**) transmit spatial angle and range dimensions; and (**c**) receive spatial angle and range dimensions.

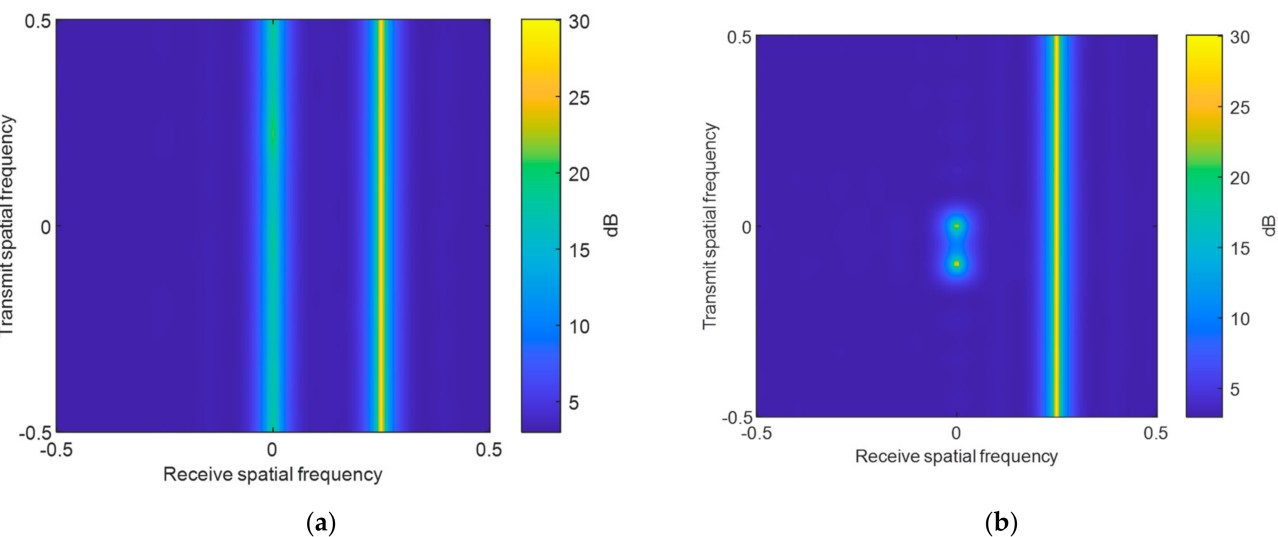

(**a**)                                                (**b**)

**Figure 5.** Capon power spectra of receive signal (**a**) before and (**b**) after SRDC.

### 4.2. Transmit–Receive Two-Dimensional Adaptive Beamforming

To verify the effectiveness of the transmit–receive two-dimensional adaptive beamforming approach, Figure 6 plots the output signal in the range and Doppler dimensions. After receive beamforming, the suppressive jamming is mitigated, resulting in the output signal in Figure 6a. However, the RGPO false targets cannot be mitigated with the receiving beamforming. Note that the RGPO false targets are randomly modulated in the Doppler domain, which is similar to that in Figure 4a. In contrast, part of RGPO false targets are well-suppressed after transmit beamforming, as show in Figure 6b. As stated previously, those false targets that are delayed to the next pulse can be well-suppressed. However, the false targets that are within same pulse as the true target cannot be suppressed at this stage. For clearance, we further plot the range profile corresponding to the Doppler frequency of true target in Figure 7. In this simulation, the moving target is in the range of 10 km, and the corresponding range gate index is 667, as indicated in Figure 7. It is seen that those false targets in front of the true target in the range dimension are well-suppressed, which is helpful for the identification of the leading edge of the true target. In this case, a possible enlarged range windowing strategy, as proposed in this paper, can be applied. It is pointed

out that the range region can be properly designed, in order to balance the computational complexity.

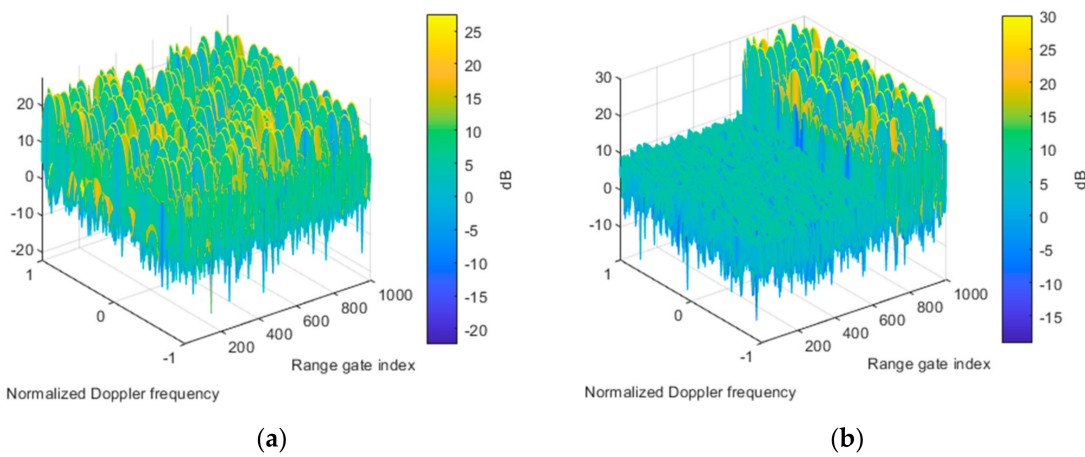

    (**a**)    (**b**)

**Figure 6.** Fourier power spectra of output signal in normalized Doppler frequency and range dimensions. (**a**) Receive beamforming. (**b**) Transmit beamforming.

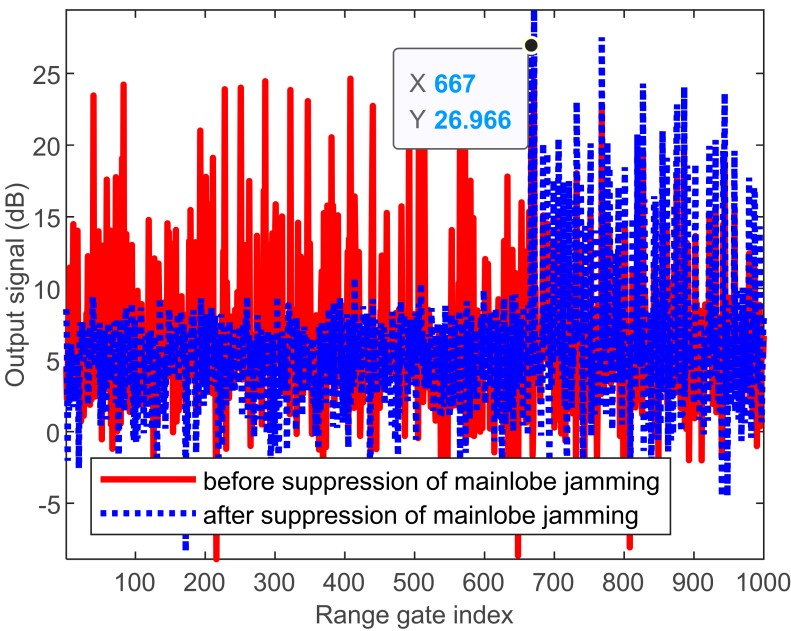

**Figure 7.** Output signal in range dimension.

As aforementioned, the transmit–receive beamforming can be designed adaptively or non-adaptively. Figure 8 plots the equivalent Capon power spectrum of the constructed covariance matrix in the transmit dimension. Notice that the performance of non-adaptive beamforming depends on the system parameters, array calibration error, and received signals. Generally, the RGPO false targets are delayed, with no more than four pulses in practice. Nevertheless, we designed many nulls of the beampattern to guarantee jamming suppression performance. Moreover, the uncertainty set associated with each null is used to widen the nulls for robustness. In this example, the uncertainty set is defined as $\xi\left[-\frac{1}{2M},\frac{1}{2M}\right]$, where $\xi$ is a ratio factor that controls the uncertainty set size. It is seen that a large uncertainty set results in the spread of power spectrum Figure 9 shows the two-dimensional transmit–receive adaptive beampattern. It is seen that the null of the beampattern is aligned to the suppressive jamming and RGPO false targets. Thus, the jamming suppression performance can be maintained.

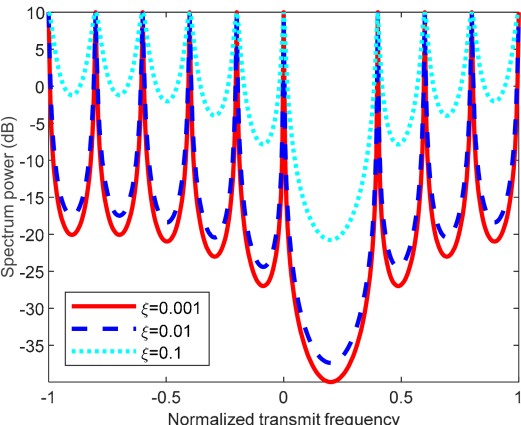

**Figure 8.** Equivalent Capon power spectrum of constructed covariance matrix in the transmit dimension.

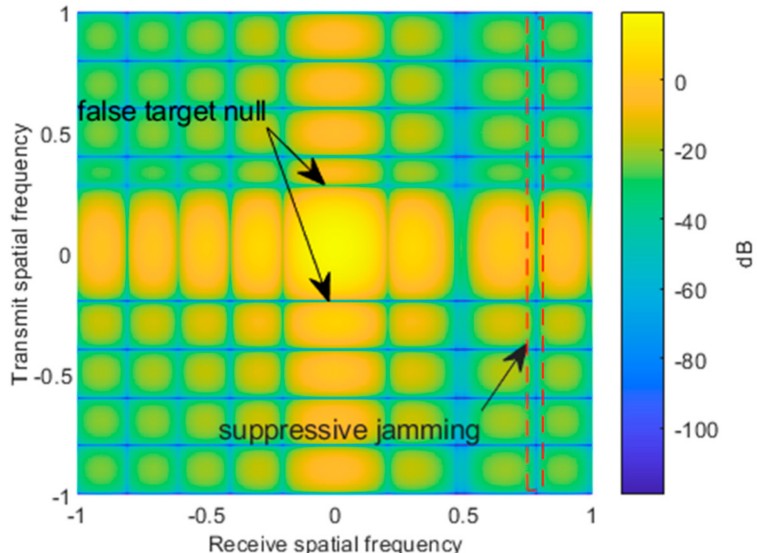

**Figure 9.** Two-dimensional transmit–receive adaptive beamforming.

### 4.3. RGPO Mainlobe Jamming Suppression Performance Analysis

The RGPO can be suppressed with the two-dimensional transmit–receive beamforming in the adaptive and non-adaptive sense. In this subsection, we testify to the performance of the RGPO mainlobe jamming suppression. Here, we define the ratio of the frequency increment to PRF as $\gamma = \frac{\Delta f}{f_{PRF}}$. It is demonstrated that the ratio should not be an integer for effective jamming suppression [34]. The output signal-to-jamming-plus-noise ratio (SJNR) is calculated, with respect to different delayed pulses and ratios of frequency increments to PRF, as presented in Figure 10. The performance of non-adaptive beamforming is plotted in Figure 10a, and the performance of adaptive beamforming is plotted in FI It is seen that the jamming suppression performance degrades dramatically for some frequency increments and delayed pulses. For example, when the delayed pulse number is 10, the same as the transmit element number, it is impossible to suppress the mainlobe RGPO false target whenever choosing the frequency increment. When the delayed pulse number is 1, all these testified frequency increment are feasible for suppression of RGPO false targets. Moreover, it can be seen that the performance of non-adaptive beamforming might degrade, due to mismatches. Therefore, the parameter should be designed to obtain better RGPO jamming suppression performance.

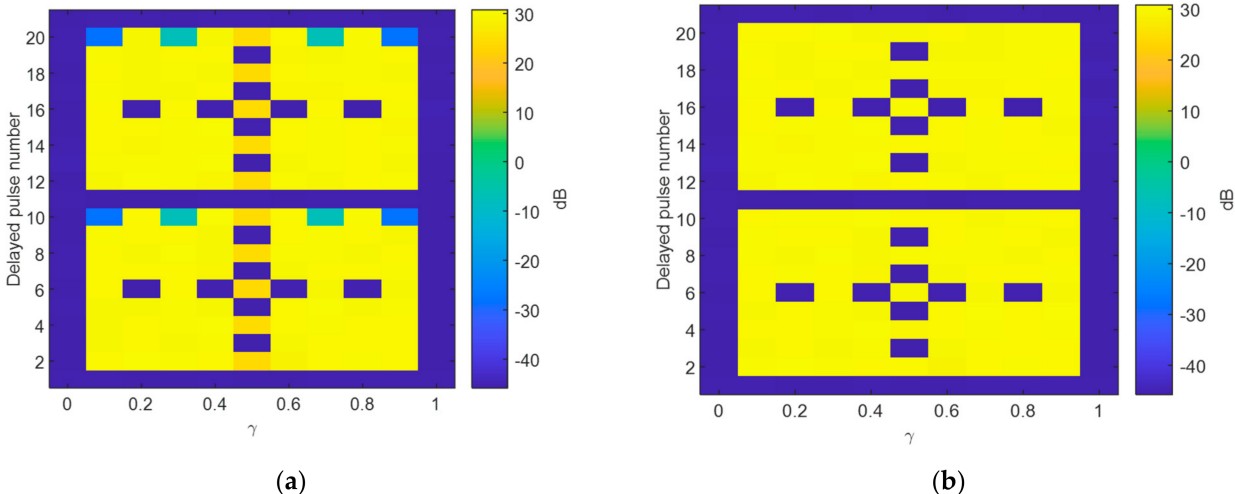

**Figure 10.** Anti-jamming performance, with respect to frequency increment and delayed pulse number. (**a**) Non-adaptive beamforming and (**b**) adaptive beamforming.

## 5. Conclusions

In this paper, a mainlobe RGPO jamming suppression approach, based on a two-dimensional transmit–receive beamforming technique, is devised by using an enlarged range windowing strategy. A series of beamformers are employed to cover the possible range region. Therefore, the true target can be abstracted with leading edge confirmation within the enlarged range region. The practical delayed time of RGPO false target are testified, and it verified that those false targets delayed to the next pulses can be suppressed effectively. Those mainlobe RGPO false targets, with small time delays, cannot be suppressed during the transmit–receive beamforming procedure. It is also verified that the true target is clear in its previous range region. The non-adaptive beamforming can be designed without the training data, and it requires proper design of the system parameters. Besides, for better RGPO mainlobe jamming suppression performance, the parameters for system design and two-dimensional adaptive beamforming should be properly designed. Future works include maintaining jamming suppression performance under non-ideal errors circumstance, non-orthogonal waveforms conditions, and practical constraints of hardware.

**Author Contributions:** Conceptualization, P.W. and J.X.; methodology, P.W. and J.X.; software, P.W.; validation, Y.W.; writing—original draft preparation, P.W.; writing—review and editing, J.X.; supervision, G.L.; funding acquisition, G.L. All authors have read and agreed to the published version of the manuscript.

**Funding:** This research was funded in part by Nature Science Foundation of China under grant numbers 61931016 and 62071344, in part by the Young Talent Starlet in Science and Technology in Shaanxi under Grant No. 13, in part by Key Laboratory Equipment Advanced Research Fund under grant number 6142206200210. The APC was funded by Nature Science Foundation of China.

**Institutional Review Board Statement:** Not applicable.

**Informed Consent Statement:** Not applicable.

**Data Availability Statement:** Please contact the Jingwei Xu for the supporting materials.

**Conflicts of Interest:** The authors declare no conflict of interest.

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
