# Peer review of "Range Gate Pull-Off Mainlobe Jamming Suppression Approach with FDA-MIMO Radar: Theoretical Formalism and Numerical Study"

_remotesensing, doi:10.3390/rs14061499_

Round 1

Reviewer 1 Report

The paper is rather well written (in term of content). The paper concent and methodology is correst. The introduction is very good and clear. 

But the paper requires corrections in many places.

  1. Some drawings showing the results of the analysis are illegible and need to be corrected (maybe enlarged, the font should be larger) - Fig.4, 5, 7.
  2. Conclusion is very short and fairly general - they should be expanded.
  3. Authors should consider introducing units to descriptions of axes, especiallly in Fig.9.
  4. Please check line 149: the variable p I think should lower case (not upper).
  5. Line 148, 153 - Equations 10 and 11. Please consider introducing spaces between variables - formlas are difficult to read.
  6. Line 160 - what is ESM? I have not found explanation for this abbreviation. 
  7. References are not written according to the guidelines.
  8. Lines: 300, 301, 306, 314, 315, 316 and maybe others - text should be rather Figure 6b than Figure 6(b).
  9. References to the literature should be placed in parentheses, for example : line 25-26 "defense applications [1,2]".
  10. Please, check instructions for authors - the paper has many editing errors.

Reviewer 2 Report

This paper discussed a main lobe RGPO jamming suppression approach based on the transmit-receive two-dimensional spatial beamforming technique. The results seem interesting, and technically sound. But the presentation can be further improved. Below are some comments:

  1. The in-text citations in this paper are quite confusing. In general, all numeric styles should follow the same system, in which a number in superscript or brackets is given as an in-text citation.
  2. The authors need to clarify the contributions clearly. In particular, the authors need to emphasize the difference compared with the existing work.
  3. The authors might consider adding one paragraph summarizing the notations.
  4. The notation system is quite confusing in this paper. First, the notation for the same variable is not consistent in the paper. For example, m and n exist in both italicized and non-italicized forms. Second, the same variable is used to represent different meanings. For example, i is used to represent the jamming signal in Eq. (10), but it is also used in (11),  (12) and (17) for a different physical meaning. The authors need to check the paper carefully for similar issues.
  5. Figure 9 is quite confusing. What is the unit in the y-axis? In addition, how is the color bar associated with the figure? More explanations are needed.

Reviewer 3 Report

Here are some comments on the paper:

line 41: why "et. al."?

line 80: 24,34,35

line 103: MIMO

line 105: "omni-directional, identical and uniform" at this stage it seems acceptable but a comment should be added, after in the paper, regarding the possible discrepancies when a real array with beamforming is used

line 138: "in anti-jammer from the mainlobe direction." at this stage it seems acceptable but a comment should be added, after in the paper, regarding the constraints on local oscillator and phase noise witch allow such a processing in a real radar

Line 196 to 206: this section could be a little bit clarified

line 208: it don't look like a power spectrum

line 253: "obtain the frontier of the interested range region and thus to protect the true target in the"  this stage it seems acceptable but a comment should be added, after in the paper, regarding the evolving of global noise level while the false targets are removed and the global power budget reduction due to the FDA-MIMO operation; the orthogonality should also be discussed

line 331: the reason why the 700-1000 section is still jammed is not clear enough

line 343: this section as well as the figure 10 is not clear at all for me. Tracking is mentioned but not detailed.

All: the link between section 4 and the previous ones is more or less clear up to line 296. The end of the paper should be more detailed.

Round 2

Reviewer 2 Report

The authors have addressed all my concerns. 

Reviewer 3 Report

The paper can be published.